# Side-Polish Plastic Optical Fiber Based SPR Sensor for Refractive Index and Liquid-Level Sensing

**DOI:** 10.3390/s22166241

**Published:** 2022-08-19

**Authors:** Chuanxin Teng, Shiyuan Ying, Rui Min, Shijie Deng, Hongchang Deng, Ming Chen, Xiaoxue Chu, Libo Yuan, Yu Cheng, Minmin Xue

**Affiliations:** 1Guangxi Key Laboratory of Optoelectronic Information Processing, School of Optoelectronic Engineering, Guilin University of Electronic Technology, Guilin 541004, China; 2State Key Laboratory of Cognitive Neuroscience and Learning, Center for Cognition and Neuroergonomics, Beijing Normal University at Zhuhai, Zhuhai 519087, China; 3School of Life and Environmental Sciences, Guilin University of Electronic Technology, Guilin 541004, China

**Keywords:** surface plasma resonance (SPR), plastic optical fiber (POF), side-polish, simultaneous measurement, refractive index (RI), liquid level

## Abstract

In this work, a simple side-polish plastic optical fiber (POF)-based surface plasmon resonance (SPR) sensor is proposed and demonstrated for simultaneous measurement of refractive index (RI) and liquid level. The effects of side-polish depths on the sensing performance were studied. The experimental results show that the SPR peak wavelength will be changed as the RI changes, and the SPR peak intensity will be changed with the liquid level variation. By monitoring the changes in peak wavelength and intensity, the RI and liquid level can be detected simultaneously. Experimental results show that an RI sensitivity of 2008.58 nm/RIU can be reached at an RI of 1.39. This sensor has the advantages of simple structure and low cost, which has a good prospect in the field of biochemical sensing.

## 1. Introduction

Surface plasmon resonance (SPR) is a physical optical phenomenon based on the interaction between incident light and free electrons in the metal layer [1,2]. As early as 1968, Otto et al. proposed an SPR excitation device with a prism as an optically coupled structure [3], and subsequently, Kretschmann and Raether further improved the Otto structure, and the metal film was directly plated on the prism surface, which can effectively excite the SPR effect [4]. Nowadays, most of the commercial SPR sensors are based on the optical prism; however, the optical prism-based SPR sensors are large in volume and complex in structure. Compared with the prism-based SPR sensors, the optical-fiber-based SPR sensors have the advantages of being compact in size, lightweight and simple in structure and having a distributed measurement ability, which can be used in real-time and in suit monitoring [5,6]. Since R.C. Jorgenson et al. first proposed an SPR sensor coupling with optical fiber in 1993 [7], it has opened the door to the research and applications of optical-fiber-based SPR sensing technology. In recent years, different kinds of optical fibers have been employed for SPR sensing, including the single-mode optical fiber [8], the multi-mode optical fiber [9], the photonic crystal fiber [10], the multicore optical fiber [11], the hollow-core optical fiber [12], the glass capillary tube [13], and so on. In order to excite the SPR, the fiber structure should be modified. However, most of the reported SPR sensors are based on glass optical fibers (GOFs). Although the GOF technology is relatively mature, and its testing equipment is relatively complete, the glass material is hard and fragile. After structural modification, the GOFs can be easily damaged. Compared with GOF, plastic optical fiber (POF) is soft and has a low melting temperature, which is easy to structurally modify [14]. Many kinds of POF structures were proposed for SPR sensing, for example the single side-polish POF-based SPR sensor [15], the double side-polish POF-based SPR sensor [16], the tapered POF-based SPR sensor [17], the U-shape POF-based SPR sensor [18], the etched POF-based SPR sensors [19], and so on. To date, the POF-based SPR sensors have been applied to chemical, biological, and many other sensing fields [20].

On the other hand, with the increasing variety of detection objects and the complexity of the detection environment, sensors that can only measure one parameter will no longer meet the needs of practical applications. In many applications, it is necessary to detect multiple parameters simultaneously, such as the refractive index (RI), temperature, liquid level, etc. Therefore, the multi-parameter SPR sensors have attracted much attention [21]. Recently, more and more research has shifted the focus to optical-fiber-based multi-parameter SPR sensors, for example, the simultaneous measurement of salinity, temperature, and pressure [22]; the simultaneous measurement of temperature and magnetic field strength [23]; the simultaneous measurement of solution concentration and temperature [24]; and the simultaneous measurement of RI and temperature [25,26,27,28]. However, to our knowledge, there are few reports about optical-fiber-based SPR sensors for simultaneous measurement of RI and liquid level. The liquid level measurement plays an important role in high-precision, real-time monitoring in the fields of fuel storage and chemical reaction, and it is very useful to monitor the RI value when measuring the liquid level at the same time. Recently, Monserrat del Carmen Alonso-Murias et al. proposed a SPR fiber tip sensor for measuring RI, temperature and level at the same time [29]. However, the measurement range is limited for the probe, and a reflecting film should be fabricated on the fiber end face, which increases the fabrication procedure. In addition, C. Teng et al. fabricated an SPR sensor based on a side-polish POF with micro-holes for simultaneous measurement of RI and liquid level [30]; however, the micro-hole structure is difficult to fabricate, and the POF will become fragile after drilling the micro-holes. Due to the surface tension, the measured liquid is also difficult to get completely inside and outside the micro holes.

In this paper, a simple POF-based SPR sensor is proposed for simultaneous measurement of RI and liquid level. The sensor probe comprises a side-polished structure, which is an easy fabrication. An obvious change in SPR peak intensity can be observed when the liquid level changes. The influence of side-polishing depth on sensor performance is studied. Experimental results show that the RI can be measured by detecting the wavelength shift of the SPR resonant peak, and the liquid level can be measured by monitoring the depth of the resonant peak. In the RI range of 1.335 to 1.39, an RI sensitivity of 2008.58 nm/RIU can be achieved. The sensor probe has the advantages of simple structure, easy fabrication, and low cost, which gives it great application prospects in the fields of biomedical sensing.

## 2. Sensor Probe Structure and Fabrication

As shown in Figure 1a, the sensing probe is a gold-coated side-polish POF; *L* represents the length of the polishing area, *d* is the polished region depth, and *D* is the residual diameter of the fiber. The cross-section view of the sensing probe is shown in Figure 1b. Except for the fiber core, the thin leftover cladding coated with a gold film may also take part in the SPR sensing. The optical fiber used in this study is a step index fiber (ESKA, CK-40) POF with a fiber diameter of 1000 μm. The core diameter of the fiber is 980 μm with an RI of 1.49 and a thermal–optical coefficient of −1.15 × 10^−4^/°C. The cladding thickness is 10 μm with a low RI of 1.41 and thermal–optical coefficient of −3.50 × 10^−4^/°C. The fabrication process for this sensor probe is shown in Figure 2. The side-polish structure is fabricated by a wheel-type fiber polishing machine [31]. A 2000-mesh sandpaper is rolled on the wheel to grind the POF repeatedly along the axis of the fiber, and the polishing parameters can be set on the computer and are used to control the movement of the wheel. In this process, both ends of the fiber need to be fixed with fiber holders. Different polishing depths can be achieved by controlling the rotation time and speed of the wheel, and finally the remaining thickness of the optical fiber is measured by a three-dimensional, real-time diameter system. The polished accuracy of the system is about ±10 μm. After this polishing process, the polished surface needs to be further smoothed. In this process, a small amount of polishing paste, composed of alumina as a main component, is squeezed onto a polishing cloth, and then the cloth is used to further polish the area for about 4–5 min until the polishing surface is smooth enough. After these processes, the remaining polishing paste on the side-polish POF is cleaned by an ultrasonic cleaning machine and dried for the gold-film coating process. A plasma sputtering device is used to deposit a layer of gold film on the side-polish area. During this process, a layer of gold film with an optimal thickness of 50 nm and accuracy of ±3 nm [28] is deposited on the polished area, and the gold film thickness is monitored using a film thickness detector. Figure 3 shows a photograph of the POF probe.

## 3. Sensor Probe Operating Principle

For the wavelength modulation mode, when *p*-polarized light in POF reaches the surface of metal film at the sensing region, part of the wave vector penetrates into the metal to form an evanescent wave. The SPR can be excited when the horizontal wave vector of evanescent wave and the surface plasma wave (SPW) satisfy the following formula [32]:$$k_{x}\;=\frac{\omega }{c}\sqrt{\varepsilon_{0\left(\lambda \right)}}\cdot \sin\theta_{spr}=k_{sp}=\frac{\omega }{c}\cdot \sqrt{\frac{\varepsilon_{1}\left(\lambda \right)\varepsilon_{2}\left(\lambda \right)}{\varepsilon_{1}\left(\lambda \right)+\varepsilon_{2}\left(\lambda \right)}}$$
where $\varepsilon_{1}$ and $\varepsilon_{2}$ are the permittivity of metal and environmental media, respectively. At this time, the incident light will be coupled to the SPW through the evanescent field, and then propagate and gradually decay, so the light intensity at the corresponding wavelength position will be reduced, resulting in an SPR peak. The SPR peak is very sensitive to the RI value of the external environment, and when the RI changes, the SPW propagation constant of the metal surface will be changed, and the propagation constant of the coupled light matched to it will also be changed, resulting in the movement of the SPR peak. Therefore, the external RI can be measured by detecting the SPR peak wavelength. When measuring the liquid level, as the solution level increases, the active length of the SPR sensing area will increase, leading to the SPR peak intensity variations, so the liquid level change can be detected by monitoring the SPR peak intensity.

The response of the proposed SPR sensor operating in wavelength interrogation mode can be simulated by using a three-layer membrane structure, namely a three-layer structural model of the core, metal, and sample. For *p*-polarized light, the normalized transmittance can be expressed as follows [29]:$$P_{trans}=\frac{\int_{\theta_{c}}^{\frac{\pi }{2}}R_{P}^{N}P\left(\theta \right)d\theta }{\int_{\theta_{c}}^{\frac{\pi }{2}}P\left(\theta \right)d\theta }$$
where $\theta_{c}=\sin^{-1}(n_{cl}/n_{co})\;$ is the critical angle of total reflection, $n_{co}$ and $n_{cl}$ are the RI of the POF core and cladding, and $P\left(\theta \right)=(n_{co}^{2}\sin\theta \cos\theta )/{\left(1-n_{co}^{2}cos^{2}\theta \right)}^{2}$ is the modal power corresponding to the angle of incidence *θ*. $R_{P}$ is the total reflectivity [29], N=L′/(2D·tanθ) is the total amount of light reflection in the sensing region, where *D* is the remaining diameter of the fiber, and *θ* is the angle of incidence at the fiber core–gold film interface. *L′* is the effective length of the sensor immersed in the measured liquid. In our simulations, for RI sensing, *L′* is 22 mm, the total polished-area length for the liquid level sensing. *L′* increases with the liquid level increases. The dielectric constant of the gold film is determined by the Drude model [33]. The thermal-light coefficient of deionized water is −8 × 10^−5^/°C [29], and $n_{co}$ determined by the Sellmeier equation [34]. In the simulation, only the meridional rays are considered.

The simulation results of the RI sensing for the probe are shown in Figure 4. The RI of the sample solution ranges from 1.335 to 1.39. As shown in Figure 4, the SPR peaks redshift as the solution RI increases, and the peak width of the SPR also increases. This means that the RI of the liquid can be detected by monitoring the SPR peak wavelength position.

Figure 5 shows the simulation results of the liquid level sensing for the probe. The RIs of the measured sample are 1.335 and 1.38. It can be seen that the SPR peak intensity decreases with the increase in the liquid level. Figure 5a shows the simulation result of the probe in the solution with an RI of 1.335, and it can be seen that the corresponding wavelength of the SPR peak is located at 510 nm, while for the measured liquid with RI of 1.38, the corresponding wavelength of the SPR peak shifts to the wavelengths of 620 nm. Therefore, the result shows that the liquid level and RI can be measured simultaneously by detecting the SPR peak intensity and wavelength, respectively. Figure 5c,d show the simulation results of liquid level sensing performance of probes with different polished depths in solutions with RIs of 1.335 and 1.38. It can be seen that the probe with a side-polished depth of 300 μm can produce the deepest SPR peak. This is because, as the polished depth increases, the number of light reflections at the SPR active region will also increase.

## 4. Experimental Results and Discussions

Figure 6 shows the schematic of experiment setup. A white light source (ideaoptics, HL2000) with a wavelength range of 360–2500 nm was used to provide the transmission light and a spectrum analyzer (ideaoptics, Shanghai, China) with a wavelength range of 325–1100 nm, and a resolution of 0.77 nm was used to detect the transmission spectrum. When the experiment was carried out, the sensor probe was placed vertically into the beaker, and the two ends of it were fixed so that the polished surface was perpendicular to the bottom of the beaker. The two ends of the fiber were connected to the light source and the spectrometer by using the fiber adapters, respectively. Deionized water with an RI of 1.335 and glycerol-aqueous solutions with RI range of 1.34–1.39 were prepared for subsequent experimental measurements. The RI of the solution was measured by the Abbe refractometer. In the RI sensing experiment, the probe was immersed in the solution; after measuring an RI solution, the sensor probe should be cleaned and dried for the next RI measurement. The experiment was performed at a temperature of 20 °C, and the SPR spectrum was obtained by normalizing the transmission spectrum into the transmission spectrum of air, and in order to obtain a smooth spectrum, 20 adjacent wavelengths’ data were averaged. For the liquid-level sensing, the transmission spectra were recorded when the beginning of the polished region was immersed by the measured liquid.

Figure 7 shows the sensing performance of the probe with a polished depth of 100 μm. As shown in Figure 7a, as the RI increases, the SPR peak of the probe undergoes a redshift [35]. Figure 7b shows the relationship of the RI and SPR peak wavelength. It can be seen that the wavelength changes nonlinearly with RI. Figure 7c,d show the liquid level sensing performance of the probe in solutions with RIs of 1.335 and 1.38, respectively. For each measurement, it can be seen that as the liquid level increases, the SPR peak intensity decreases. It can also be seen that the SPR peak position is different for the measured liquid with a different RI. For the deionized water with an RI of 1.335, the SPR peak was generated at the wavelength of 580 nm, and in the solution with an RI of 1.38, the SPR peak is located at the wavelength of 640 nm. The experiment results clearly demonstrate that the probe is capable of measuring RI and liquid level simultaneously, and the experiment results are similar to the simulation results, except for some differences in SPR peak wavelength position and width. The difference between the experimental and simulation results may be caused by the following aspects: there are a lot of light rays propagating in the fiber, including the meridional rays and skew rays, but only the meridional rays were considered in the simulation. Another reason may be the non-uniformity of the probe structure, including the non-uniformity of polished structure, gold-film thickness, and even the liquid film in the immediate vicinity of the gold layer. Figure 7e,f show the relationships of the liquid level and the SPR peak intensity for the probe in the solutions with RIs of 1.335 and 1.38, respectively. It can be seen that the SPR peak intensity changes nonlinearly with the liquid level.

In order to study the effect of polished depth on the sensing performance, the probes with polished depths of 200 and 300 μm were fabricated and tested, and the polished length was also 22 mm. Figure 8a,b show the RI sensing performances for the probes with polished depths of 200 and 300 μm, respectively. The SPR peaks of the two probes are redshifted with the increase in RI. Figure 8c shows the relationship between the SPR peak wavelength and the RI for the probes with polished depths of 100, 200, and 300 μm, respectively. After a polynomial fitting, the sensitivity for each probe at different RI points is shown in Figure 8d. The results show that in the RI range of 1.335–1.38, the probe with a polished depth of 100 μm has a higher sensitivity, but the probe with a polished depth of 200 μm achieves the maximum sensitivity of 2008.58 nm/RIU when the solution RI is 1.39. The sensitivity can be determined by S=*dλ*/*dn*, where *S* is the sensitivity, *dλ* is the change in wavelength, and *dn* is the change in liquid RI. Figure 8e shows the transmission spectra of three probes with different polished depths of 100, 200, and 300 μm in deionized water. It can be seen that the probe with a 200 μm polished depth has the narrowest full-width at half-maximum (FWHM), while the probe with a polished depth of 300 μm has the deepest SPR peak. Figure 8f shows the FWHMs of the transmission spectrum for the three probes in different RI sample solutions. The results show that in the RI range of 1.335–1.38, the FWHM corresponding to the probe with a polished depth of 200 μm is narrower than those of the other two probes.

The liquid-level sensing performance of the probes with different polished depths is shown in Figure 9. Figure 9a,b show the relationship between the liquid level and SPR peak intensity in the solution with RIs of 1.335 and 1.38, respectively. It can be seen that for all the probes, the normalized transmission changes nonlinearly with the liquid level. The result shows that the peak intensity decreases obviously when the liquid level is low, and as the liquid level increases, the peak intensity changes slightly, which has a similar trend with the simulation results. It can be also seen that the SPR peak intensity is the deepest for the probe with a polished depth of 300 μm.

In addition, the temperature influence of the sensor was also tested. The probe with a polished depth of 200 μm was used; the measurement sample was deionized water and the temperature changed from 24 to 32 °C with increments of 2 °C. Figure 10a shows the transmission spectra for the probe at different temperatures, ranging from 24 to 32 °C. Figure 10b shows the corresponding changes in SPR peak wavelength position and intensity at different temperatures. It can be seen that the SPR peak intensity fluctuates between 0.676 to 0.688, while the peak wavelength decreases from 604.6 to 602.8 nm as the temperature increases from 24 to 32 °C. The changes of SPR peak wavelength may be induced by the combined influence of the thermo-optic effects of POF itself, and deionized water. Additionally, the changes in peak intensity may be mainly caused by the fluctuation of the light source.

## 5. Conclusions

A simple side-polish, POF-based SPR sensor was proposed and implemented for simultaneous RI and level measurement in this paper. The probe consisted of a side-polish structure, and a gold layer was coated on the polished surface. A three-layer structural model was used to analyze the sensor performance. The simulation results showed that the probe was able to achieve simultaneous measurement of RI and liquid levels. As the RI increased, the SPR peak wavelength redshifted, and as the liquid level increased, the SPR peak intensity decreased. The experimental results were similar to the simulation results. In order to test the effects of polished depth on the sensing performance, the sensing probes with a side-polish length of 22 mm and depths of 100, 200, and 300 μm were fabricated. Results showed that the sensitivity and FWHM will be different for the probe with different side-polish depths. An RI sensitivity of 2008.58 nm/RIU can be achieved at an RI of 1.39 for the probe with a polished depth of 200 μm, and the SPR peak is the deepest for the probe with a polished depth of 300 μm. Finally, the temperature influence of the sensor was tested in the temperature range of 24–32 °C. The probe has potential applications in biosensing fields and has the advantages of easy fabrication, simple structure, and low cost.

## Figures and Tables

**Figure 1 sensors-22-06241-f001:**
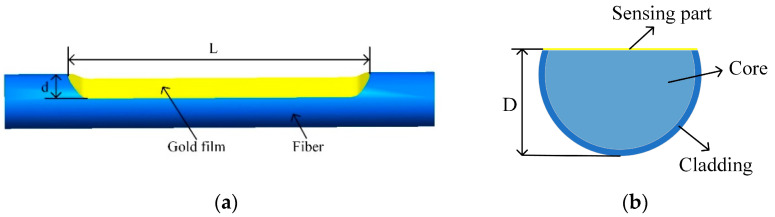
(**a**) Schematic of the side-polish POF sensing probe; (**b**) The cross-section view of the side-polish POF sensing probe.

**Figure 2 sensors-22-06241-f002:**
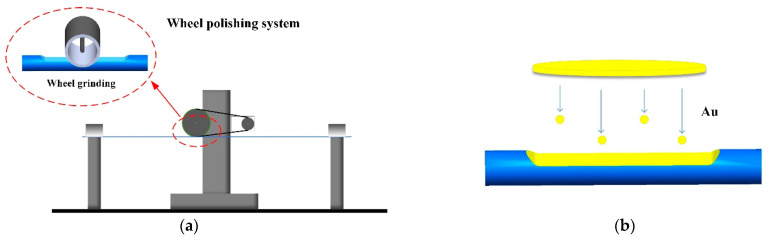
Schematic of the fabrication process for the sensor probe: (**a**) Side-polish process; (**b**) Sputtering process.

**Figure 3 sensors-22-06241-f003:**
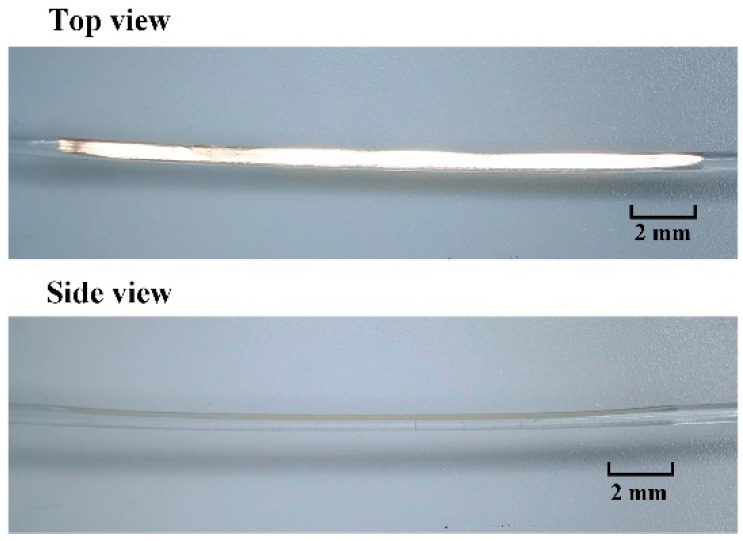
Photograph of the POF probe.

**Figure 4 sensors-22-06241-f004:**
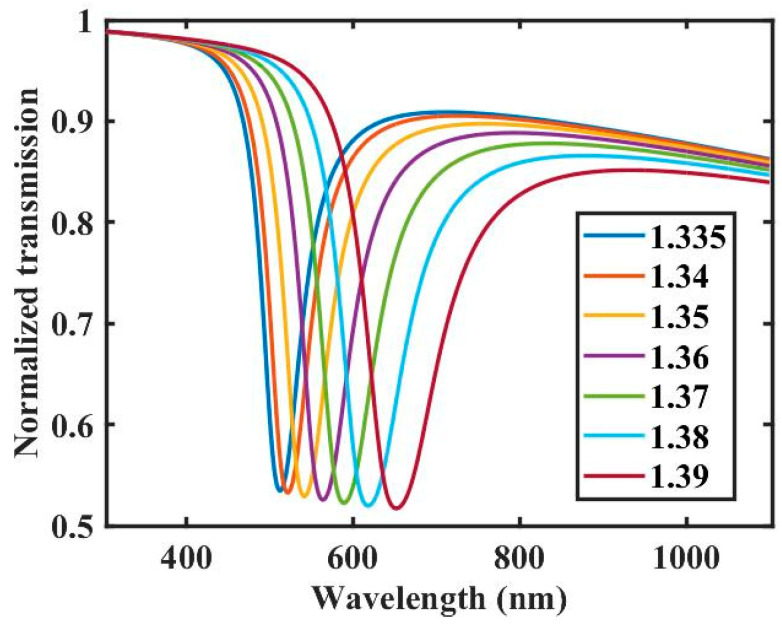
Simulation results of RI sensing.

**Figure 5 sensors-22-06241-f005:**
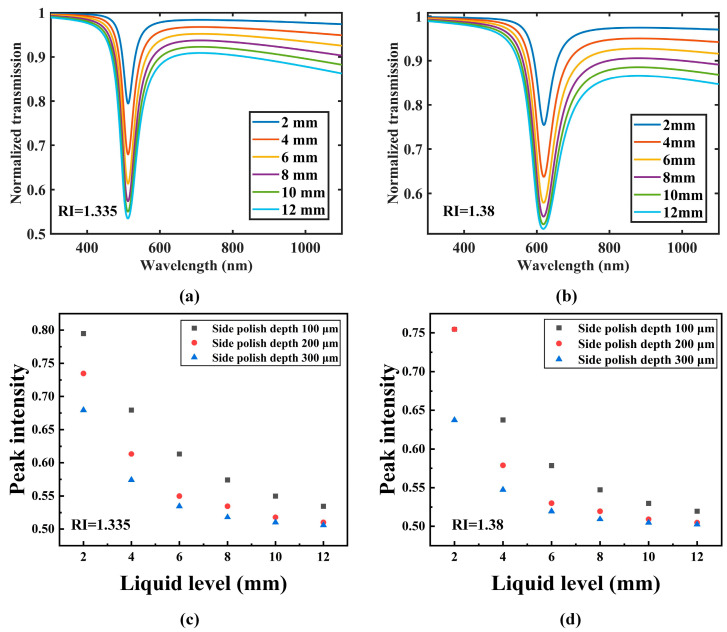
Simulation results of the liquid level sensing: (**a**) The solution with RI of 1.335; (**b**) The solution with RI of 1.38; The simulation results of liquid level sensing performance of probes with different polished depths in the liquid with RIs of 1.335 (**c**) and 1.38 (**d**).

**Figure 6 sensors-22-06241-f006:**
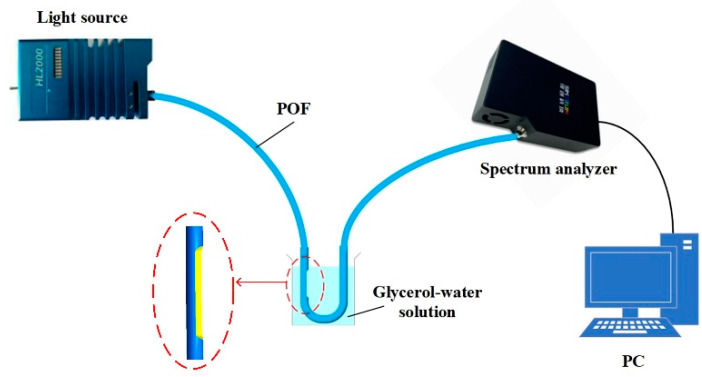
The schematic of experiment setup.

**Figure 7 sensors-22-06241-f007:**
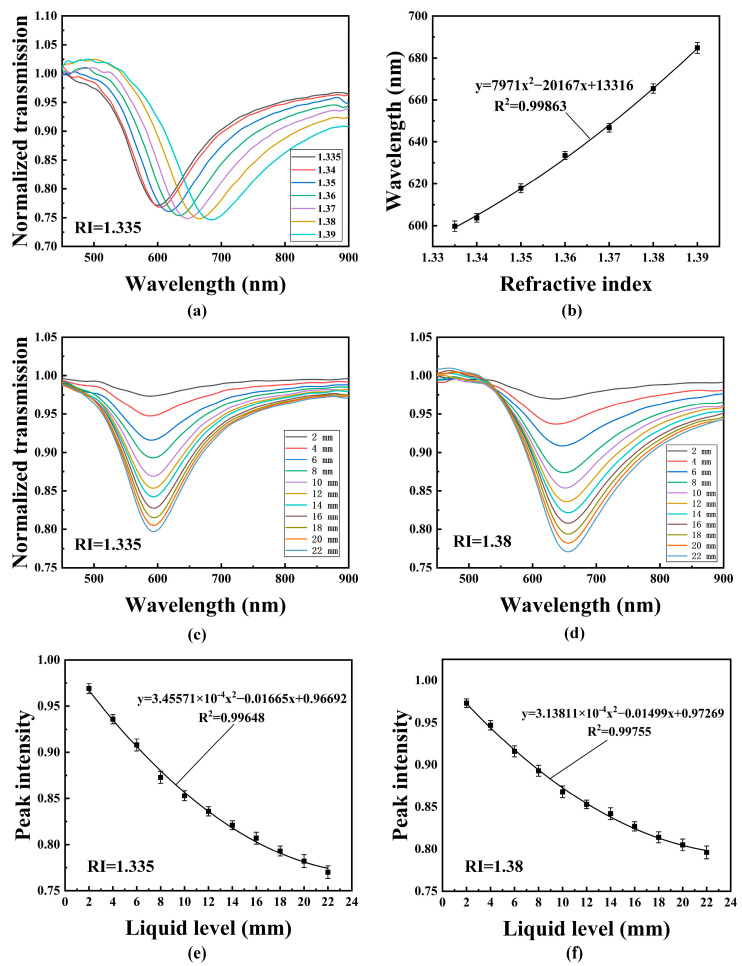
The sensing performance of the POF probe with polished depth of 100 μm: (**a**) The transmission spectrum for RI measurement; (**b**) The relationship between RI and SPR peak wavelength; The transmission spectrum for liquid level measurement with RI of 1.335 (**c**) and 1.38 (**d**); The relationship between liquid level and SPR peak intensity with RI of 1.335 (**e**) and 1.38 (**f**).

**Figure 8 sensors-22-06241-f008:**
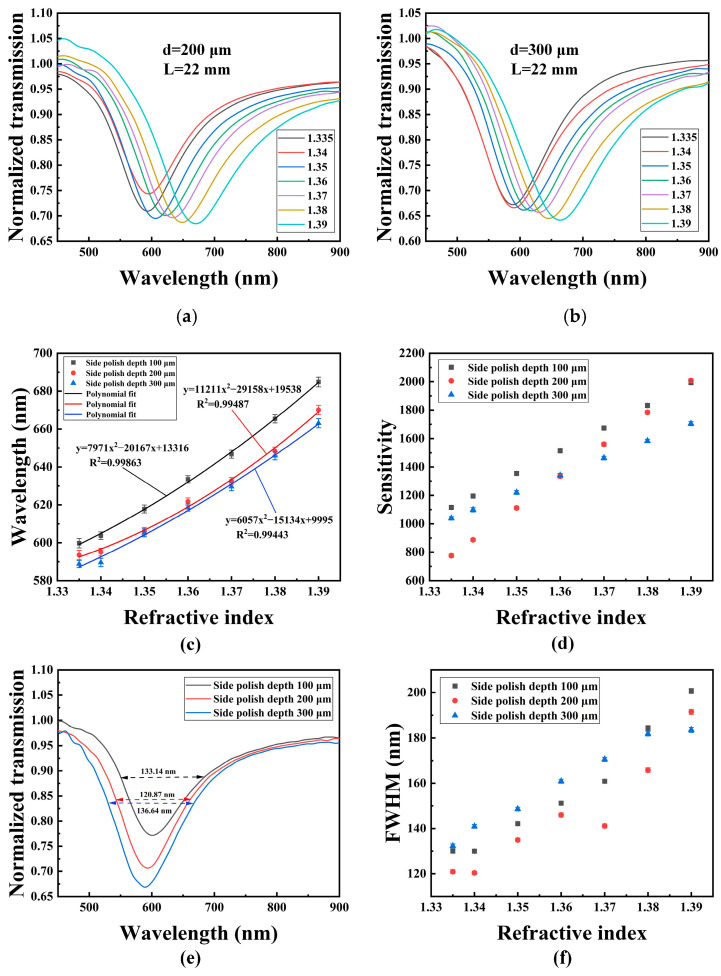
RI sensing performance for the POF probes with different depths: (**a**) 200 μm; (**b**) 300 μm; (**c**) The relationship of SPR peak wavelength and RI for probes with different polished depths; (**d**) RI sensitivity of probes with different polished depth; (**e**) Normalized transmission spectra of the POF probes with different polished depths in water; (**f**) FWHM of probes with different polished depth in different samples.

**Figure 9 sensors-22-06241-f009:**
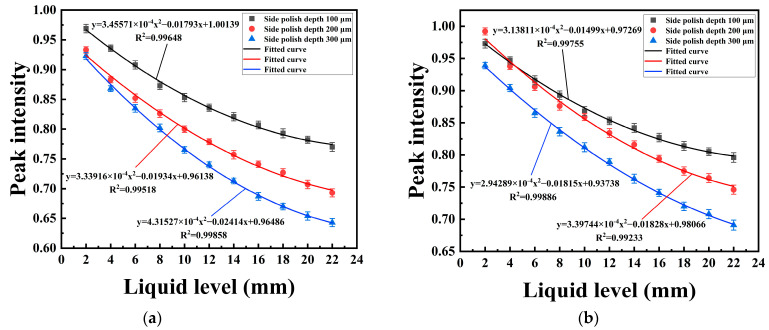
Liquid-level sensing performance of probes with different polished depths in liquids with RIs of 1.335 (**a**) and 1.38 (**b**).

**Figure 10 sensors-22-06241-f010:**
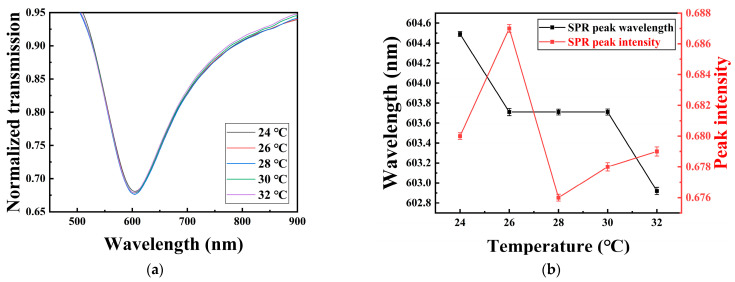
(**a**) Normalized transmission spectra for the probe with different temperatures ranging from 24 to 32 °C; (**b**) The changes in SPR peak wavelengths and transmission intensities at different temperatures.

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
