# Peer review of "Side-Polish Plastic Optical Fiber Based SPR Sensor for Refractive Index and Liquid-Level Sensing"

_sensors, 2022, doi:10.3390/s22166241_

Round 1

Reviewer 1 Report

·       In this manuscript, Teng et al. demonstrate a simple side-polish plastic optical fiber (POF) based surface plasmon resonance (SPR) sensor is proposed and demonstrated for simultaneous measurement of refractive index (RI) and liquid level. The manuscript is interesting, it is in general deserves publication in Sensors. My comments below will improve the content of the manuscript. My comments are as follows:
1. The manuscript English must be improved in the revised version of the manuscript.
2. The novelty of the manuscript should be clearly highlighted.
3. Are these optimized results? The authors should use the optimized results.
4. What type of resonance causes the polished depth on the sensing performance. I suggest that the author add theoretical explanations to the following questions
Liquid level sensing performanceRI sensing performance for the POF probes with different depths.
5. The author should include a few recent references in the manuscript as follows:

“Amplitude and frequency tunable absorber in the terahertz range”, Results in Physics, 2022, 34:105263“Wave-thermal effect of a temperature-tunable terahertz absorber”, Optics Express, 2021, 29(23): 38557-38566.“Analyzing broadband tunable metamaterial absorber by using symmetry model”, Optics Express, 2021, 29(25): 41475-41484

Reviewer 2 Report

The paper presents a simple and inexpensive optical sensor based on plastic optical fiber. The original idea was to measure both the refractive index and the level of the liquid by a POF sensor. The paper is well written and organized. The current state of the art and literature overview of the topic are presented meticulously and purposefully.

The paper contains a simplified theory of plasmonic interaction, which suffices for the paper. The experimental results are extensive and complete.

Paper shortages and errors.

There is no information on measurement errors. The errors determine the sensor sensitivity and should be provided in the paper. Furthermore, measurement errors could explain the results in Fig. 10b.

The authors claim that differences between theory and experiment presented in Fig. 5 and Fig. 7, are caused by non-meridional modes of multimode fiber. This is not true, as different modes can result in different plasmonic absorption intensities; however, the location of the peak does not change. The differences are probably caused by the nonuniformity of the liquid film in the immediate vicinity of the gold layer.

The illustration should be added, showing geometry of the top view of the polished fiber, with explanations of which part of the structure is available for SPR sensing. Note: exposed core obviously takes part in the sensing; however, parts with thin leftovers of cladding can also take part in the sensing.

Reviewer 3 Report

The authors proposed a simple side-polish plastic optical fiber (POF) based surface plasmon resonance (SPR) sensor. This work is well written and the results are interesting for the readers. However, a minor revision is needed to accommodate the high-quality requirements of this Journal. 

1.    The thickness of gold is 50 nm in the proposed structure. Is the thickness of 50 nm of gold an optimal value? It should have an available range. Therefore, the influence of the thickness of the gold-film coating on the transmittance spectrum should be briefly described in the text.

2.   The simulation method should be elucidated in the text. For example, if you used COMSOL Multiphysics, the simulation setting using COMSOL should be clarified in more detail, e.g., the version of COMSOL, the simulation model, 2-D or 3-D, mesh size, permittivity of gold and dielectric medium, PML or SBC boundary.

3.      On page 6, it states “As shown in Figure 7a, as the RI increases, the SPR peak of the probe undergoes a redshift.” The mechanism of “redshift” should quote related literature (see Scientific reports, 2021, 11(1), 18515).

4.      On page 6, it states “the reason for the differences between the experimental and simulation results may be that there are a lot of light rays propagating in the fiber, including the meridional rays and skew rays, but only the meridional rays were considered in the simulation.” In addition to the abovementioned reason, I think that the discrepancy between measured and simulated results is inevitable because of the uniformity of the structure. I suggest including this point in the revised manuscript.

5.      In the introduction section, please show the novelty points of the current work and compare it with the other optical sensor with RI sensing, e.g., J. Nanopart. Res., 2020, 22(9), 297 and J. Phys. D: Appl. Phys., 2021, 54(11), 115301.
